# DYNAMIC WEIGHTED PROJECTION MAINTENANCE WITH $\ell_p$-LEWIS WEIGHT

## ABSTRACT

We introduce a new data–structure problem—*Dynamic $\ell_p$-Lewis Weight Projection Maintenance*—that asks us to maintain the projection

$$P(W) = W^{1/2-1/p} A(A^\top W^{1-2/p}A)^{-1} A^\top W^{1/2-1/p}$$

under a stream of diagonal weight updates and to support fast matrix–vector products with $P(W)$. This setting strictly generalizes the $\sqrt{W}A$ projection, which is at the heart of state-of-the-art linear programming and interior point methods, and it captures a wide range of algorithms that rely on leverage scores or Lewis weights for sampling and preconditioning. We provide a deterministic projection-maintenance data structure with sublinear amortized updates. Moreover, we extend it to the differential privacy setting.

## 1 INTRODUCTION

Projection maintenance is one of the most important data structure problems in modern convex optimization, serving as a critical component in achieving the best-known runtime guarantees for many cutting-edge algorithms (Lee et al., 2019; Jiang et al., 2020b;a; Cohen et al., 2021b; Huang et al., 2022). We first recall the definition of classical Dynamic Projection Maintenance Problem.

**Definition 1.1** (Dynamic Projection Maintenance (Cohen et al., 2021b)). *Given a matrix $B \in \mathbb{R}^{m \times n}$, the goal is to design a data structure to maintain the projection matrix $P(B) := B(B^\top B)^{-1}B^\top$ and support the fast multiplication of $P(B) \cdot h$ for any query $h \in \mathbb{R}^n$ with the following operations:*

- INIT($B \in \mathbb{R}^{m \times n}$): *The data structure takes the matrix $B$ as input, and does some preprocessing and compute an initial projection.*

- UPDATE($B^{\mathrm{new}} \in \mathbb{R}^{m \times n}$): *The data structure receives some low-rank or sparse update $B^{\mathrm{new}}$ and updates $B$ by $B + B^{\mathrm{new}}$.*

- QUERY($h \in \mathbb{R}^n$): *The data structure receives a vector $h$ and approximately computes the matrix-vector product of updated projection matrix $P(B)$ and an online vector $h$.*

For example, in linear programming (Cohen et al., 2021b), we take $B = \sqrt{W}A$, where $A$ is the constraint matrix and $W$ is a diagonal matrix representing slack variables. In each iteration, $W$ undergoes relatively small perturbations. The goal of the data structure is to efficiently approximate

$$\sqrt{W}A(A^\top WA)^{-1}A^\top \sqrt{W}h$$

for an online vector $h \in \mathbb{R}^n$.

In this work, we consider a specific projection maintenance problem with $B = W^{1/2-1/p}A$ which generalizes the above problem. We need to maintain the projection and compute an approximation of matrix-vector product between the projection matrix and any online vector $h \in \mathbb{R}^n$. We call this problem, for maintaining such kind of matrices, the Dynamic $\ell_p$-Lewis Weight Projection Maintenance. Formally, it is defined as follows.

**Definition 1.2** (Dynamic $\ell_p$-Lewis Weight Projection Maintenance). *Given $p > 0$, a matrix $A \in \mathbb{R}^{m \times n}$ and a diagonal matrix $W \in \mathbb{R}^{m \times m}$ with nonnegative entries, the goal is to design a data structure to maintain the projection matrix*

$$P(W) := W^{1/2-1/p}A(A^\top W^{1-2/p}A)^{-1}A^\top W^{1/2-1/p}$$

*and support the fast multiplication of $P(W) \cdot h$ for any query $h \in \mathbb{R}^n$ with the following operations:*

- INIT($A \in \mathbb{R}^{m \times n}, W \in \mathbb{R}^{m \times m}$): *The data structure takes a matrix $A \in \mathbb{R}^{m \times n}$ and a diagonal matrix $W \in \mathbb{R}^{m \times m}$ with nonnegative entries as input, and does some preprocessing and compute an initial projection.*

- UPDATE($W^{\mathrm{new}} \in \mathbb{R}^{m \times m}$): *The data structure receives some low-rank or sparse update $W^{\mathrm{new}}$ and updates $W$ by $W + W^{\mathrm{new}}$.*

- QUERY($h \in \mathbb{R}^n$): *The data structure receives a vector $h$ and approximately computes the matrix-vector product of updated projection matrix $P(W)$ and an online vector $h$.*

This problem is fundamental to the design of efficient algorithms in settings where leverage scores or Lewis weights determine adaptive sampling (Cohen & Peng, 2015; Parulekar et al., 2021; Brand et al., 2021a; Woodruff & Yasuda, 2023) or preconditioning (Durfee et al., 2018; Yang et al., 2018; Kyng et al., 2019).

**Roadmap.** In Section 2, we review relevant literature related to our study. In Section 3, we present serveral useful tools and provide the main result. In Section 4, we state the main result of this paper. In Section 5, we provide technical overview of our study. In Section 6, we draw our conclusion.

## 2 RELATED WORK

### 2.1 LINEAR PROGRAMMING AND SEMIDEFINITE PROGRAMMING

Linear programming is a cornerstone of optimization and theoretical computer science. Dantzig's Simplex algorithm (Dantzig, 1951) remains a practical workhorse despite its exponential worst-case complexity. The Ellipsoid Method later gave the first polynomial-time guarantee for linear programming, although it is typically slower in practice than Simplex. A decisive breakthrough came with Karmarkar's interior-point method (Karmarkar, 1984), which combines polynomial running time with strong empirical performance and has sparked extensive work on ever-faster interior-point techniques for a broad range of optimization problems (Vaidya, 1987; Renegar, 1988; Vaidya, 1989; Daitch & Spielman, 2008; Lee & Sidford, 2013; 2014; 2019; Cohen et al., 2021a; Lee et al., 2019; Brand, 2020; Brand et al., 2020; Jiang et al., 2021; Song & Yu, 2021; Gu & Song, 2022).

Beyond their algorithmic importance, linear programming and semidefinite programming are ubiquitous in machine-learning theory. They underpin efficient formulations for empirical risk minimization (Lee et al., 2019; Song et al., 2022b; Qin et al., 2023), support vector machines (Gu et al., 2023; Gao et al., 2023a), and numerous other learning problems, providing both rigorous guarantees and scalable implementations.

### 2.2 SKETCHING

Sketching—compressing data with random linear maps—has become a workhorse across modern optimization and numerical linear algebra. It powers cutting-edge algorithms for linear programming (Jiang et al., 2021; Song & Yu, 2021), empirical risk minimization (Lee et al., 2019; Qin et al., 2023), and semidefinite programming (Jiang et al., 2020a; Huang et al., 2022; Song et al., 2023c). In randomized numerical linear algebra it accelerates a wide range of matrix tasks and decompositions (Clarkson & Woodruff, 2017; Nelson & Nguyên, 2013; Boutsidis et al., 2016; Razenshteyn et al., 2016; Song et al., 2017; Xiao et al., 2018; Song et al., 2019; Lee et al., 2019; Jiang et al., 2021; Song & Yu, 2021; Brand et al., 2021b; Hu et al., 2022; Song et al., 2022a; Gu & Song, 2022). Most applications employ *oblivious* sketches—data-independent projections—for dimensionality reduction (Clarkson & Woodruff, 2017; Nelson & Nguyên, 2013). For approximate John-ellipsoid

computation, Chen et al.(Cohen et al., 2019) rely exclusively on sketching, suggesting room for further acceleration, while Mahabadi et al.(Makarychev et al., 2022) tackle a tougher streaming variant for convex polytopes; their method, however, is not yet optimal in our computational model.

## 2.3 Differential Privacy

Introduced by (Dwork et al., 2006), differential privacy (DP) has become the gold standard for rigorous data protection. An extensive body of work now retrofits classical algorithms, data structures, and machine-learning pipelines with provable DP guarantees (Esfandiari et al., 2022; Andoni et al., 2023; Cherapanamjeri et al., 2023; Cohen-Addad et al., 2022; Dong et al., 2024; Farhadi et al., 2022; Gopi et al., 2023; Li et al., 2022; Gopi et al., 2022; Huang & Yi, 2021; Jung et al., 2019; Li & Li, 2024; Epasto et al., 2024; Chen et al., 2022; Beimel et al., 2022; Narayanan, 2022; 2023; Fan & Li, 2022; Fan et al., 2024; Li & Li, 2023; Eliáš et al., 2020; Yu et al., 2024; Liang et al., 2024; Gu et al., 2024; Song et al., 2023b; Qin et al., 2022; Song et al., 2023a; Galli et al., 2024; Chen et al., 2024; Romijnders et al., 2024; Qi et al., 2024; Ke et al., 2025; Hu et al., 2024; Liu et al., 2024). Beyond integrating privacy into existing methods, researchers are refining the fundamental DP building blocks themselves. Enhanced variants of the Gaussian, Exponential, and Laplace mechanisms now deliver tighter accuracy–privacy trade-offs than the classical formulations (Dwork et al., 2014). A prime example is the truncated Laplace mechanism of Gopi et al. (Geng et al., 2020), which currently achieves the smallest known error for any $(\epsilon, \delta)$-DP distribution.

## 3 Preliminary

**Fact 3.1** ((Woodbury, 1950)). *The Woodbury matrix identity is*
$$(M + UCV)^{-1} = M^{-1} - M^{-1}U(C^{-1} + VM^{-1}U)^{-1}VM^{-1}.$$

*Let $S \subset [n]$ denote the set of coordinates that is changed by more than a constant factor and $r = |S|$. Using the identity above, we have that*
$$M_{w^{\text{new}}} = M_w - (M_w)_S(\Delta_{S,S}^{-1} + (M_w)_{S,S})^{-1}((M_w)_S)^\top,$$
*where $\Delta = \text{diag}(w^{\text{new}} - w)$, $(M_w)_S \in \mathbb{R}^{n \times r}$ is the $r$ columns from $S$ of $M_w$ and $(M_w)_{S,S}, \Delta_{S,S} \in \mathbb{R}^{r \times r}$ are the $r$ rows and columns from $S$ of $M_w$ and $\Delta$.*

**Fact 3.2.** *We have*

- *Let $A \in \mathbb{R}^{n \times n}$, then we have $\|A\|_F \leq \sqrt{n}\|A\|$.*

- *Let $A \in \mathbb{R}^{n \times n}$, then we have $\|A\| \leq \|A\|_F$*

- *For two vectors $a, b \in \mathbb{R}^n$, then we have $|ab^\top| \leq \|a\|_2 \cdot \|b\|_2$*

**Definition 3.3** (Differential Privacy, (Dwork et al., 2014)). *For $\epsilon > 0, \delta \geq 0$, a randomized function $\mathcal{A}$ is $(\epsilon, \delta)$-differentially private ($(\epsilon, \delta)$-DP) if for any two neighboring datasets $X \sim X'$, and any possible outcome of the algorithm $S \subset$ in Range($\mathcal{A}$), $\Pr[\mathcal{A}(X) \in S] \leq e^\epsilon \Pr[\mathcal{A}(X') \in S] + \delta$.*

**Lemma 3.4** (Truncated Laplace Mechanism, (Dwork et al., 2014; Geng et al., 2020; Andoni et al., 2023)). *Let $\text{Lap}(\lambda)$ denote the Laplace distribution with parameter $\lambda$ with PDF $\Pr[Z = z] = \frac{1}{2\lambda}e^{-|z|/\lambda}$. Let $B_L := (\Delta/\epsilon)\log(1 + \frac{e^\epsilon - 1}{2\delta})$. Let $\text{TLap}(\Delta, \epsilon, \delta)$ denote the Truncated Laplace distribution with PDF proportional to $e^{-|z|/\lambda}$ on the region $[-B_L, B_L]$. Given a numeric function $f$ that takes a dataset $X$ as the input, and has sensitivity $\Delta$, the mechanism output $f(X) + Z$ where $Z \sim \text{Lap}(\Delta/\epsilon)$ is $(\epsilon, 0)$-DP. In addition, if $Z \sim \text{TLap}(\Delta, \epsilon, \delta)$, then $f(X) + Z$ is $(\epsilon, \delta)$-DP.*

**Definition 3.5** (Dataset, (Gao et al., 2023b)). *Fix $\eta > 0, \alpha > 0$. We say our dataset $X \in \mathbb{R}^{n \times d}$ is $(\alpha, \eta)$-good if $XX^\top \succeq \eta \cdot I_n$ and for all $i \in [d]$, $\|X_{*,i}\|_2 \leq \alpha$.*

**Definition 3.6** ($\beta$-close neighbor dataset, (Gao et al., 2023b)). *Let $B > 0$ be a constant. Let $n$ be the number of data points. Let dataset $\mathcal{D} = \{(x_i, y_i)\}_{i=1}^n$, where $x_i \in \mathbb{R}^d$ and $\|x_i\|_2 \leq B$ for any $i \in [n]$. We define $\mathcal{D}'$ as a neighbor dataset with one data point replacement of $\mathcal{D}$. Without loss of generality, we have $'\mathcal{D} = \{(x_i, y_i)\}_{i=1}^{n-1} \cup \{(x_n', y_n)\}$. Namely, we have $\mathcal{D}$ and $\mathcal{D}'$ only differ in the $n$-th item.*

*Additionally, we assume that $x_n$ and $x_n'$ are $\beta$-close. Namely, we have*
$$\|x_n - x_n'\|_2 \leq \beta.$$

**Lemma 3.7** (Post-Processing Lemma for DP, (Dwork et al., 2014)). *Let $\mathcal{M} := \mathbb{N}^{|\chi|} \to \mathbb{R}$ be a randomized algorithm that is $(\epsilon, \delta)$-differentially private. Let $f : \mathbb{R} \to \mathbb{R}'$ be an arbitrarily random mapping. Then is $f \circ \mathcal{M} : \mathbb{N}^{|\chi|} \to \mathbb{R}'$ $(\epsilon, \delta)$-differentially private.*

**Theorem 3.8** (Empirical covariance estimator for Gaussian (Vershynin, 2018)). *Let $\Sigma \in \mathbb{R}^{d \times d}$ be PSD, $X_1, \cdots, X_n \sim \mathcal{N}(0, \Sigma)$ be i.i.d and $\widetilde{\Sigma} = \frac{1}{n} \sum_{i=1}^n X_i X_i^\top$. Then with probability $1 - \gamma$, it holds that $\|\Sigma^{-1/2}\widetilde{\Sigma}\Sigma^{-1/2} - I\|_F \le \rho$ for some $\rho = O(\sqrt{\frac{d^2 + \log(1/\gamma)}{n}} + \frac{d^2 + \log(1/\gamma)}{n})$.*

**Lemma 3.9** (Composition lemma for DP, (Dwork et al., 2014)). *Let $\mathcal{M} := \mathbb{N}^{|\chi|} \to \mathbb{R}$ be an $(\epsilon_i, \delta_i)$-DP algorithm for $i \in [k]$. If $\mathcal{M}_{[k]} \to \Pi_{i=1}^n \mathcal{R}_i$ satisfies $\mathcal{M}_{[k]}(x) = (\mathcal{M}_1(x), \cdots, \mathcal{M}_k(x))$, then $\mathcal{M}_{[k]}$ is $(\sum_{i=1}^k \epsilon_i, \sum_{i=1}^k \delta_i)$-DP.*

**Lemma 3.10** (Lemma C.15 in (Song & Yu, 2021)). *Let $x^{\mathrm{new}} = x + \widetilde{\delta}_x$ and $s^{\mathrm{new}} = s + \widetilde{\delta}_s$. Let $w = \frac{x}{s}$ and $w^{\mathrm{new}} = \frac{x^{\mathrm{new}}}{s^{\mathrm{new}}}$. Then we have $\sum_{i=1}^n (\mathbb{E}[\ln w_i^{\mathrm{new}}] - \ln w_i)^2 \le 64\epsilon^2$, $\sum_{i=1}^n (\mathrm{Var}[\ln w_i^{\mathrm{new}}])^2 \le 1000\epsilon^2$.*

# 4 MAIN RESULT

The goal of this section is to prove the following theorem:

---

**Algorithm 1** Projection Maintenance Data Structure

---

1: **datastructure** MAINTAINPROJECTION
2:
3: **members**
4:     $w \in \mathbb{R}^n$
5:     $v, \widetilde{v} \in \mathbb{R}^n$
6:     $A \in \mathbb{R}^{d \times n}$
7:     $M \in \mathbb{R}^{n \times n}$
8:     $Q \in \mathbb{R}^{n \times n^b L}$
9:     $R_{1,*}, R_{2,*}, \cdots, R_{L,*} \in \mathbb{R}^{n^b \times n}$
10:     $l \in \mathbb{N}_+, L \in \mathbb{N}_+$
11:     $\epsilon_{\mathrm{mp}} \in (0, 1/4)$
12:     $a \in (0, \alpha]$
13: **end members**
14:
15: **procedure** INITIALIZE($A, w, \epsilon_{\mathrm{mp}}, a$)
16:     $w \leftarrow w, v \leftarrow w, \epsilon_{\mathrm{mp}} \leftarrow \epsilon_{\mathrm{mp}}, A \leftarrow A, a \leftarrow a$
17:     $M \leftarrow A^\top (AV^{1-2/p}A^\top)^{-1}A$
18:     Choosing $R_{1,*}, R_{2,*}, \cdots, R_{L,*} \in \mathbb{R}^{n^b \times n}$ to be sketching matrices
19:     $R \leftarrow [R_{*,1}, R_{*,2}, \cdots, R_{*,L}]$
20:     $Q \leftarrow MV^{1/2-1/p}R^\top$
21:     $l \leftarrow 1$
22: **end procedure**
23:
24: **end datastructure**

---

**Theorem 4.1** (Projection maintenance). *Given a full rank matrix $A \in \mathbb{R}^{d \times n}$ with $n \ge d$ and a tolerance parameter $0 < \epsilon_{\mathrm{mp}} < 1/4$. Given any positive number $a$ such that $a \le \alpha$ where $\alpha$ is the dual exponent of matrix multiplication. Let $R_{1,*}, \cdots, R_{L,*} \in \mathbb{R}^{n^b \times n}$ denote a list of sketching matrices, where $b \in [0, 1]$. There is a deterministic data structure (Algorithm 1) that approximately maintains the projection matrices*

$$W^{1/2-1/p}A^\top (AW^{1-2/p}A^\top)^{-1}AW^{1/2-1/p}$$

*for positive diagonal matrices $W$ through the following two operations:*

*1. UPDATE($w$): Output a vector $\widetilde{v}$ such that for all $i \in [n]$,*

$$(1 - \epsilon_{\mathrm{mp}})\widetilde{v}_i^{1/2-1/p} \le w_i^{1/2-1/p} \le (1 + \epsilon_{\mathrm{mp}})\widetilde{v}_i^{1/2-1/p}.$$

2. QUERY($h$): *Output* $\widetilde{V}^{1/2-1/p}A^\top(A\widetilde{V}^{1-2/p}A^\top)^{-1}A\widetilde{V}^{1/2-1/p}(R^\top)_{*,l}R_{l,*}h$ *for the $\widetilde{v}$ outputted by the last call to* UPDATE.

*The data structure takes $n^2d^{\omega-2}$ time to initialize and each call of QUERY($h$) takes time $n^{1+b+o(1)}+n^{1+a+o(1)}$.*

*Furthermore, if the initial vector $w^{(0)}$ and the (random) update sequence $w^{(1)},\cdots,w^{(T)}\in\mathbb{R}^n$ satisfies*

$$\sum_{i=1}^n\big((1/2-1/p)\cdot(\mathbb{E}[\ln w_i^{(k+1)}]-\ln w_i^{(k)})\big)^2\le C_1^2 \text{ and } \sum_{i=1}^n\mathrm{Var}[(1/2-1/p)\ln w_i^{(k+1)}])^2\le C_2^2$$

*with the expectation and variance is conditional on $w_i^{(k)}$ for all $k=0,1,\cdots,T-1$. Then, the amortized expected time[1] per call of UPDATE($w$) is $(C_1/\epsilon_{\mathrm{mp}}+C_2\epsilon_{\mathrm{mp}}^2)\cdot(n^{\omega-1/2+o(1)}+n^{2-a/2+o(1)})$.*

*Proof.* The theorem holds by combining Lemma 4.3, Lemma 4.4 and Lemma 4.5. □

**Remark 4.2.** *For our linear program algorithm, we have $C_1=O(1/\log n)$, $C_2=O(1/\log n)$ and $\epsilon_{\mathrm{mp}}=\Theta(1)$. See Lemma 3.10.*

To verify the correctness of our updates, we have the following lemma:

**Lemma 4.3** (Correctness of the algorithm, informal version of Lemma C.1)**.** *The output of* UPDATE($w$) *in Algorithm 2 satisfies*

$$M=A^\top(AV^{1-2/p}A^\top)^{-1}A,\ Q=MV^{1/2-1/p}R^\top$$

*The output of* QUERY($h$) *in Algorithm 3 satisfies*

$$p_s=\widetilde{P}(R^\top)_{*,l}R_{l,*}h$$
$$p_x=(I-\widetilde{P})(R^\top)_{*,l}R_{l,*}h,$$

*where $\widetilde{P}=V^{1/2-1/p}A^\top(A\widetilde{V}^{1-2/p}A^\top)^{-1}A\widetilde{V}^{1/2-1/p}$, and $\widetilde{V}$ is outputted by* UPDATE($w$).

Above lemma verifies our algorithm. Now we consider the running time of the projection maintenance, which consists of Initialization time, update time and query time, as discussed below.

## 4.1 INITIALIZATION TIME, UPDATE TIME

To formalize the amortized runtime proof, we first analyze the initialization time (Lemma 4.4), update time (Lemma 4.5), and query time (Lemma 4.6) of our projection maintenance data-structure.

**Lemma 4.4** (Initialization time)**.** *The initialization time of data-structure* MAINTAINPROJECTION *(Algorithm 1) is $O(n^2d^{\omega-2})$.*

*Proof.* Given a matrix $A\in\mathbb{R}^{d\times n}$ and diagonal matrix $V\in\mathbb{R}^{n\times n}$, computing $A^\top(AVA^\top)^{-1}A$ takes $O(n^2d^{\omega-2})$. □

**Lemma 4.5** (Update time)**.** *The update time of data-structure* MAINTAINPROJECTION *(Algorithm 2) is $O(rg_rn^{2+o(1)})$ where $r$ is the number of indices we updated in $v$.*

*Proof.* The proof is identical to (Cohen et al., 2021b; Lee et al., 2019). We omit the details here. □

## 4.2 QUERY TIME

**Lemma 4.6** (Query time, informal version of Lemma C.2)**.** *The query time of data-structure* MAINTAINPROJECTION *(Algorithm 1) is $O(n^{1+b+o(1)}+n^{1+a+o(1)})$.*

---

[1]If the input is deterministic, so is the output and the runtime.

**Algorithm 2** Update

1: **datastructure** MAINTAINPROJECTION
2:
3: **procedure** UPDATE($w$)
4:     $y_i \leftarrow \ln w_i - \ln v_i, \forall i \in [n]$
5:     $r \leftarrow$ the number of indices $i$ such that $|y_i| \geq \epsilon_{\mathrm{mp}}/2$.
6:     **if** $r < n^a$ **then**
7:         $v^{\mathrm{new}} \leftarrow v$
8:         $M^{\mathrm{new}} \leftarrow M$
9:         $l \leftarrow l + 1$
10:     **else**
11:         Let $\pi : [n] \to [n]$ be a sorting permutation such that $|y_{\pi(i)}| \geq |y_{\pi(i+1)}|$
12:         **while** $1.5 \cdot r < n$ and $|y_{\pi(\lceil 1.5 \cdot r\rceil)}| \geq (1 - 1/\log n)|y_{\pi(r)}|$ **do**
13:             $r \leftarrow \min(\lceil 1.5 \cdot r\rceil, n)$
14:         **end while**
15:         $v^{\mathrm{new}}_{\pi(i)} \leftarrow \begin{cases} w_{\pi(i)} & i \in \{1, 2, \cdots, r\} \\ v_{\pi(i)} & i \in \{r+1, \cdots, n\} \end{cases}$
16:         $\Delta \leftarrow \mathrm{diag}(v^{\mathrm{new}} - v)$
17:         $\Gamma \leftarrow \mathrm{diag}((v^{\mathrm{new}})^{1/2-1/p} - v^{1/2-1/p})$
18:         Let $S \leftarrow \pi([r])$ be the first $r$ indices in the permutation.
19:         Let $M_S \in \mathbb{R}^{n \times r}$ be the $r$ columns from $S$ of $M$.
20:         Let $M_{S,S}, \Delta_{S,S} \in \mathbb{R}^{r \times r}$ be the $r$ rows and columns from $S$ of $M$ and $\Delta$.
21:         $M^{\mathrm{new}} \leftarrow M - M_{*,S} \cdot (\Delta^{-1}_{S,S} + M_{S,S})^{-1} \cdot (M_{*,S})^{\top}$
22:         Re-generate $R$
23:         $Q^{\mathrm{new}} \leftarrow Q + (M^{\mathrm{new}} \cdot \Gamma) \cdot R^{\top} + (M^{\mathrm{new}} - M) \cdot V^{1/2-1/p} \cdot R^{\top}$
24:         $l \leftarrow 1$
25:     **end if**
26:     $v \leftarrow v^{\mathrm{new}}$
27:     $M \leftarrow M^{\mathrm{new}}$
28:     $Q \leftarrow Q^{\mathrm{new}}$
29:     $\widetilde{v}_i \leftarrow \begin{cases} v_i & \text{if } |\ln w_i - \ln v_i| < \epsilon_{\mathrm{mp}}/2 \\ w_i & \text{otherwise} \end{cases}$
30:     **return** $\widetilde{v}$
31: **end procedure**
32:
33: **end datastructure**

**Algorithm 3** Query

1: **datastructure** MAINTAINPROJECTION
2:
3: **procedure** QUERY($h$)
4:     Let $\widetilde{S}$ be the indices $i$ such that $|\ln w_i - \ln v_i| \geq \epsilon_{\mathrm{mp}}/2$.
5:     $\widetilde{\Delta} \leftarrow \widetilde{V}^{1-2/p} - V^{1-2/p}$
6:     $\widetilde{\Gamma} \leftarrow \widetilde{V}^{1/2-1/p} - V^{1/2-1/p}$
7:     $p_m \leftarrow \widetilde{V}^{1/2-1/p} \cdot (M_{*,\widetilde{S}}) \cdot (\widetilde{\Delta}^{-1}_{\widetilde{S},\widetilde{S}} + M_{\widetilde{S},\widetilde{S}})^{-1} \cdot (Q_{\widetilde{S},l} + M_{\widetilde{S},*} \cdot \widetilde{\Gamma} \cdot (R^{\top})_{*,l}) \cdot R_{l,*} \cdot h$
8:     $p_s \leftarrow \widetilde{V}^{1/2-1/p} \cdot (Q_{*,l} + M \cdot \widetilde{\Gamma} \cdot (R^{\top})_{*,l}) \cdot R_{l,*} \cdot h - p_m$
9:     $p_x \leftarrow (R^{\top})_{*,l} \cdot R_{l,*} \cdot h - p_s$
10:     **return** $(p_x, p_s)$
11: **end procedure**
12:
13: **end datastructure**

## 5 TECHNICAL OVERVIEW

In this section, we present technical overview of our study. In Section 5.1, we introduce the key parameters for privacy analysis. In Section 5.2, we analyze the DP guarantees for $W^{1/2-1/p}A$, while Section 5.3 investigates its utility guarantees. In Section 5.4, we present DP guarantees for $A^\top W^{1/2-1/p}$. In Section 5.5, we present utility guarantees for $A^\top W^{1/2-1/p}$. In Section 5.6, we provide DP guarantees for $(A^\top W^{1-2/p}A)^{-1}$. In Section 5.7, we provide utility guarantees for $(A^\top W^{1-2/p}A)^{-1}$.

### 5.1 KEY CONCEPTS

**Definition 5.1** (Definition of $M$, (Gao et al., 2023b), see Definition 5.1 of (Gu et al., 2025) as an example)**.** *Let $\mathcal{M} : (\mathbb{R}^n)^d \to \mathbb{R}^{n \times n}$ be a (randomized) algorithm that given a dataset of $d$ points in $\mathbb{R}^n$ outputs a PSD matrix. Let $\mathcal{Y}, \mathcal{Y}' \in (\mathbb{R}^n)^d$. Then, we define*

$$M := \|\mathcal{M}(\mathcal{Y})^{1/2}\mathcal{M}(\mathcal{Y}')^{-1}\mathcal{M}(\mathcal{Y})^{1/2} - I\|_F.$$

**Definition 5.2** (Definition of $\Delta$, (Gao et al., 2023b), see Definition 5.2 of (Gu et al., 2025) as an example)**.** *If we have the following conditions:*

- *Let $\epsilon \in (0, 1)$ and $\delta \in (0, 1)$.*

- *Let $k$ denote the number of i.i.d. samples $g_1, g_2, \cdots, g_k$ from $\mathcal{N}(0, \Sigma_1)$ output by Algorithm 4.*

*We define $\Delta := \min \left\{ \frac{\epsilon}{\sqrt{8k \log(1/\delta)}}, \frac{\epsilon}{8 \log(1/\delta)} \right\}.$*

### 5.2 DP GUARANTEES FOR $W^{1/2-1/p}A$

**Lemma 5.3** (Sensitivity of $W^{1/2-1/p}A$, informal version of Lemma D.1)**.** *If the following conditions hold:*

- *Let the neighboring dataset $X$ and $X'$ be defined in Definition 3.6.*

- *Let $\epsilon_J, \delta_J \in \mathbb{R}$ denote the DP parameters for $J$.*

- *Let $J := W^{1/2-1/p}A$ denote the data generated by $X$ and $J'$ denote the data generated by neighboring dataset $X'$, where $W^{1/2-1/p} \in \mathbb{R}^{m \times m}$ and $A \in \mathbb{R}^{m \times n}$.*

- *Let $\beta > 0$ be defined as Definition 3.6.*

*Then, we can show that the sensitivity of $J$ is $\sqrt{n} \cdot \beta$.*

Then, we use the truncated Laplace mechanism (Lemma 3.4) to ensure the DP property of $W^{1/2-1/p}A$.

**Lemma 5.4** (DP guarantees for $W^{1/2-1/p}A$)**.** *If the following conditions hold:*

- *Let the neighboring dataset $X$ and $X'$ be defined in Definition 3.6.*

- *Let $\epsilon_J, \delta_J \in \mathbb{R}$ denote the DP parameters for $J$.*

- *Let $\Delta_J := \sqrt{n} \cdot \beta$ denote the sensitivity of $J$.*

- *Let $J := W^{1/2-1/p}A$ denote the data generated by $X$ and $J'$ denote the data generated by neighboring dataset $X'$, where $W^{1/2-1/p} \in \mathbb{R}^{m \times m}$ and $A \in \mathbb{R}^{m \times n}$.*

- *Let $\beta > 0$ be defined as Definition 3.6.*

- *Let $B_L = (\Delta_J/\epsilon_J) \log(1 + \frac{\exp(\epsilon_J)-1}{2\delta_J})$.*

- *Let $\widetilde{J} := J + \text{TLap}(\Delta_J, \epsilon_J, \delta_J)$.*

*Then, we can show that $\widetilde{J}$ is $(\epsilon_J, \delta_J)$-DP.*

*Proof.* The proof follows directly from Lemma 3.4. □

### 5.3 UTILITY GUARANTEES FOR $W^{1/2-1/p}A$

**Lemma 5.5** (Utility guarantees for $W^{1/2-1/p}A$, informal version of Lemma E.1)**.** *Under the same conditions in Lemma 5.4, we can show that $\|\widetilde{J} - J\|_2 \leq \sqrt{n} \cdot B_L$.*

### 5.4 DP GUARANTEES FOR $A^\top W^{1/2-1/p}$

**Lemma 5.6** (DP guarantees for $A^\top W^{1/2-1/p}$)**.** *If the following conditions hold:*

- *Let the neighboring dataset $X$ and $X'$ be defined in Definition 3.6.*

- *Let $\epsilon_J, \delta_J \in \mathbb{R}$ denote the DP parameters for $J$.*

- *Let $\Delta_J := \sqrt{n} \cdot \beta$ denote the sensitivity of $J$.*

- *Let $J^\top := A^\top W^{1/2-1/p}$ denote the data generated by $X$ and $J'^\top$ denote the data generated by neighboring dataset $X'$, where $W^{1/2-1/p} \in \mathbb{R}^{m \times m}$ and $A^\top \in \mathbb{R}^{n \times m}$.*

- *Let $\beta > 0$ be defined as Definition 3.6.*

- *Let $B_L = (\Delta_J/\epsilon_J) \log(1 + \frac{\exp(\epsilon_J)-1}{2\delta_J})$.*

- *Let $\widetilde{J^\top} := J^\top + \text{TLap}(\Delta_J, \epsilon_J, \delta_J)$.*

*Then, we can show that $\widetilde{J^\top}$ is $(\epsilon_J, \delta_J)$-DP.*

*Proof.* In Lemma 5.4, we prove the differential privacy property of $W^{1/2-1/p}A$. By the post-processing property of differential privacy (Lemma 3.7), the transpose matrix $A^\top W^{1/2-1/p}$ computed from the privatized matrix $W^{1/2-1/p}A$ also preserves $(\epsilon_J, \delta_J)$-differentially private. □

### 5.5 UTILITY GUARANTEES FOR $A^\top W^{1/2-1/p}$

**Lemma 5.7** (Utility guarantees for $A^\top W^{1/2-1/p}$, informal version of Lemma E.2)**.** *Under the same conditions in Lemma 5.6,*

*we can show that $\|\widetilde{J^\top} - J^\top\|_2 \leq \sqrt{n} \cdot B_L$.*

### 5.6 DP GUARANTEES FOR $(A^\top W^{1-2/p}A)^{-1}$

**Lemma 5.8** (DP guarantees for $(A^\top W^{1-2/p}A)^{-1}$, Theorem 6.12 in (Gao et al., 2023b), Theorem 5.1 in (Alabi et al., 2023), Lemma 5.4 in (Gu et al., 2025), informal version of Lemma B.1)**.** *Under the same conditions in Lemma B.1, there exists an Algorithm 4 such that*

- *Part 1. Algorithm 4 is $(\epsilon_\alpha, \delta_\alpha)$-DP.*

- *Part 2. Outputs $\widehat{\Sigma} \in \mathbb{S}_+^n$ denotes the private version of input $\Sigma$, such that with probabilities at least $1 - \gamma$, $\|\Sigma^{-1/2}\widehat{\Sigma}\Sigma^{-1/2} - I_n\|_F \leq \rho$.*

- *Part 3. $(1-\rho)\Sigma \preceq \widehat{\Sigma} \preceq (1+\rho)\Sigma$.*

*By the post-processing property of differential privacy (Lemma 3.7), the inverse $\widehat{\Sigma}^{-1}$ computed from the privatized matrix $\widehat{\Sigma}$ remains $(\epsilon_\alpha, \delta_\alpha)$-differentially private.*

In Lemma 5.8, **Part 1** claims the privacy guarantees of the "Gaussian Sampling Mechanism", **Part 2** establishes the critical properties necessary to ensure the utility of the "Gaussian Sampling Mechanism", and **Part 3** presents the ultimate utility outcomes of the algorithm.

Note that in our setting, we use $\Sigma = H$, where $H$ is the non-private matrix of interest, and we also have $\widehat{\Sigma} = \widetilde{H}$ to denote the private version of $H$.

The quantity $M$ used in **Condition 6** is formally analyzed in Section A.1.

---

**Algorithm 4** The Gaussian Sampling Mechanism, (Gao et al., 2023b)

---

1: **procedure** ALGORITHM($\Sigma, k$)
2:     PSD matrix $\Sigma \in \mathbb{R}^{n \times n}$ and parameter $k \in \mathbb{N}$
3:     Obtain vectors $g_1, g_2, \cdots, g_k$ by sampling $g_i \sim \mathcal{N}(0, \Sigma)$, independently for each $i \in [k]$
4:     Compute $\widehat{\Sigma} = \frac{1}{k} \sum_{i=1}^{k} g_i g_i^\top$           ▷ Covariance estimate.
5:     **return** $\widehat{\Sigma}$
6: **end procedure**

---

### 5.7 UTILITY GUARANTEES FOR $(A^\top W^{1-2/p} A)^{-1}$

**Lemma 5.9** (Utility guarantees for $(A^\top W^{1-2/p} A)^{-1}$, informal version of Lemma E.3). *Under the same conditions in Lemma 5.8, with probability $1 - \gamma$, we have $\|H^{-1} - \widetilde{H}^{-1}\| \leq O(\frac{\rho \cdot \eta_{\max}}{\eta_{\min}^2})$.*

### 5.8 DP GUARANTEES FOR $W^{1/2-1/p} A (A^\top W^{1-2/p} A)^{-1} A^\top W^{1/2-1/p} \cdot h$

**Lemma 5.10** (DP guarantees for $W^{1/2-1/p} A (A^\top W^{1-2/p} A)^{-1} A^\top W^{1/2-1/p} \cdot h$, informal version of Lemma D.2). *If the following conditions hold:*

- *Let $\epsilon_\alpha, \delta_\alpha \in \mathbb{R}$ denote the DP parameter for $A^\top W^{1-2/p} A$.*

- *Let $\epsilon_J, \delta_J \in \mathbb{R}$ denote the DP parameters for $W^{1/2-1/p} A$ and $A^\top W^{1/2-1/p}$.*

- *Let $\epsilon = 2\epsilon_J + \epsilon_\alpha$, $\delta = 2\delta_J + \delta_\alpha$.*

- *Let $H$ and $\widetilde{H}$ be defined as Lemma 5.9.*

- *Let $J$ and $\widetilde{J}$ be defined as Lemma 5.4.*

*Then, we can show $W^{1/2-1/p} A (A^\top W^{1-2/p} A)^{-1} A^\top W^{1/2-1/p} \cdot h$ is $(\epsilon, \delta)$-DP.*

### 5.9 UTILITY GUARANTEES FOR $W^{1/2-1/p} A (A^\top W^{1-2/p} A)^{-1} A^\top W^{1/2-1/p} \cdot h$

**Lemma 5.11** (Utility guarantees for $W^{1/2-1/p} A (A^\top W^{1-2/p} A)^{-1} A^\top W^{1/2-1/p} \cdot h$, informal version of Lemma E.4). *Under the same conditions as in Lemma E.4, with probability $1 - \gamma$, we have*

$$|JH^{-1}J^\top \cdot h - \widetilde{J}\widetilde{H}^{-1}\widetilde{J}^\top \cdot h| \leq 2\sigma_J \sigma_h \sqrt{n} \cdot B_L + \sigma_J^2 \sigma_h \cdot O(\frac{\rho \cdot \eta_{\max}}{\eta_{\min}^2}).$$

## 6 CONCLUSION

In this work, we introduce *Dynamic $\ell_p$-Lewis Weight Projection Maintenance*, which is a novel data-structure that considers the projection maintenance problem $P(B) := B(B^\top B)^{-1} B^\top$ with $B = W^{1/2-1/p} A$, that strictly generalizes the $B = \sqrt{W} A$ projection. Our deterministic algorithm supports fast updates and queries with sublinear amortized time and extends naturally to the differential privacy setting with provable utility guarantees. This work not only advances theoretical tools for linear programming, interior point methods, and leverage-based algorithms, but also opens avenues for private and efficient optimization in data-sensitive applications.

## ETHIC STATEMENT

This paper does not involve human subjects, personally identifiable data, or sensitive applications. We do not foresee direct ethical risks. We follow the ICLR Code of Ethics and affirm that all aspects of this research comply with the principles of fairness, transparency, and integrity.

## REPRODUCIBILITY STATEMENT

We ensure reproducibility of our theoretical results by including all formal assumptions, definitions, and complete proofs in the appendix. The main text states each theorem clearly and refers to the detailed proofs. No external data or software is required.

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

# Appendix

**Roadmap.**

- In Section A, we provide a sensitivity analysis and perturbation bounds for PSD Matrices in DP guarantees.

- In Section B, we introduce the Gaussian Sampling Mechanism.

- In Section C, we present a detailed proof of the main result stated in Section 4.

- In Section D, we provide detailed proof of DP guarantees introduced in Section 5.

- In Section E, we give a complete proof of the utility guarantees outlined in Section 5.

## A    SENSITIVITY AND SPECTRAL PERTURBATION OF PSD MATRIX

### A.1    SENSITIVITY OF PSD MATRIX

In this section, we provide more lemmas related to sensitivity.

**Lemma A.1** (Lemma D.1 in (Gao et al., 2023b))**.** *If $X \in \mathbb{R}^{n \times d}$ and $X' \in \mathbb{R}^{n \times d}$ are neighboring datasets (see Definition 3.5 and Definition 3.6), then $(1 - 2\alpha\beta/\eta)XX^\top \preceq X'X'^\top \preceq (1 + 2\alpha\beta/\eta)XX^\top$.*

*Proof.* Let $i \in [d]$ be index that $X_{*,i}$ and $X'_{*,i}$ are different (See Definition 3.6).

We have

$$
\begin{aligned}
X'X'^\top &= \sum_{j=1}^{d} X'_{*,j}X'^{\top}_{*,j} \\
&= (\sum_{j \in [d]\backslash\{i\}} X'_{*,j}X'^{\top}_{*,j}) + X'_{*,i}X'^{\top}_{*,i} \\
&= (\sum_{j \in [d]\backslash\{i\}} X_{*,j}X^{\top}_{*,j}) + X'_{*,i}X'^{\top}_{*,i} \\
&= XX^\top - X_{*,i}X^{\top}_{*,i} + X'_{*,i}X'^{\top}_{*,i}
\end{aligned}
$$

where the first step is the result of matrix multiplication, the second step is from simple algebra, the third step follows from Definition 3.6, and the last step comes from simple algebra.

We know that

$$
\begin{aligned}
\|X_{*,i}X^{\top}_{*,i} - X'_{*,i}X'^{\top}_{*,i}\| &= \|X_{*,i}X^{\top}_{*,i} - X_{*,i}X'^{\top}_{*,i} + X_{*,i}X'^{\top}_{*,i} - X'_{*,i}X'^{\top}_{*,i}\| \\
&\leq \|X_{*,i}X^{\top}_{*,i} - X_{*,i}X'^{\top}_{*,i}\| + \|X_{*,i}X'^{\top}_{*,i} - X'_{*,i}X'^{\top}_{*,i}\| \\
&\leq \|X_{*,i}\|_2 \cdot \|X_{*,i} - X'_{*,i}\|_2 + \|X_{*,i} - X'_{*,i}\|_2 \cdot \|X'_{*,i}\|_2 \\
&\leq 2\alpha\beta
\end{aligned}
\tag{1}
$$

where the first step is from adding a new term $X_{*,i}X'^{\top}_{*,i}$, the second step follows from the triangle inequality, the third step follows from Fact 3.2, and the last step is due to Definition 3.5 and Definition 3.6.

Thus, we have $X'X'^\top \succeq XX^\top - 2\alpha\beta I_n \succeq (1 - 2\alpha\beta/\eta)XX^\top$. Similarly, we have $X'X'^\top \preceq XX^\top + 2\alpha\beta I_n \preceq (1 + 2\alpha\beta/\eta)XX^\top$. □

**Lemma A.2** (Lemma D.2 in (Gao et al., 2023b))**.** *If the following conditions hold,*

- *Let $\alpha$ and $\eta$ be defined in Definition 3.5.*

- *Let $\beta$ be defined in Definition 3.6.*

- *By Lemma A.1, $X$ and $X'$ are neighboring datasets such that*

$$(1 - 2\alpha\beta/\eta)XX^\top \preceq X'X'^\top \preceq (1 + 2\alpha\beta/\eta)XX^\top$$

*Then, we have*

- *Part 1.*

$$\|(XX^\top)^{-1/2}X'X'^\top(XX^\top)^{-1/2} - I\| \leq 2\alpha\beta/\eta$$

- *Part 2.*

$$\|(XX^\top)^{-1/2}X'X'^\top(XX^\top)^{-1/2} - I\|_F \leq 2\sqrt{n}\alpha\beta/\eta$$

*Proof.* The proof is straightforward, and we omit the details here. □

**Lemma A.3** ( Spectral norm of $H - \widetilde{H}$)**.** *If we have the below conditions,*

- **Condition 1.** *If $\mathcal{D} \in \mathbb{R}^{n \times d}$ and $\mathcal{D}' \in \mathbb{R}^{n \times d}$ are neighboring dataset (see Definition 3.6)*

- **Condition 2.** *Let $H := A^\top W^{1-2/p}A$ denotes the symmetric positive semi-definite matrix generated by $\mathcal{D}$.*

- **Condition 3.** *Let $\widetilde{H}$ denote the private $H$ generated by Algorithm 4 with $H$ as the input.*

- **Condition 4.** *Let $\eta_{\max}I_{n \times n} \succeq H \succeq \eta_{\min}I_{n \times n}$, for some $\eta_{\max}, \eta_{\min} \in \mathbb{R}$.*

- **Condition 5.** *Let $\rho = O(\sqrt{(n^2 + \log(1/\gamma))/k} + (n^2 + \log(1/\gamma))/k)$.*

- **Condition 6.** *Let $\gamma \in (0, 1)$.*

*Then, with probability $1 - \gamma$, we have*

$$\|H - \widetilde{H}\| \leq \rho \cdot \eta_{\max}$$

*Proof.* By Part 3 of Lemma 5.8, with probability $1 - \gamma$, we have

$$(1 - \rho)H \preceq \widetilde{H} \preceq (1 + \rho)H$$

which implies

$$-\rho H \preceq \widetilde{H} - H \preceq \rho H \tag{2}$$

Then, we have

$$\|\widetilde{H} - H\| \leq \rho \cdot \eta_{\max}$$

□

**Lemma A.4** ((Wedin, 1973), Theorem 1.1 in (Meng & Zheng, 2010))**.** *Given two matrices $A, B \in \mathbb{R}^{d_1 \times d_2}$ with full column rank, we have*

$$\|A^\dagger - B^\dagger\| \lesssim \max(\|A^\dagger\|^2, \|B^\dagger\|^2) \cdot \|A - B\|.$$

# B   GAUSSIAN SAMPLING MECHANISM

In this section, we restate the analysis for "Gaussian Sampling Mechanism", which guarantees the privacy of our algorithm and provides potential tools for demonstrating its utility.

**Lemma B.1** (DP guarantees for $(A^\top WA)^{-1}$, Theorem 6.12 in (Gao et al., 2023b), Theorem 5.1 in (Alabi et al., 2023), Lemma D.1 in (Gu et al., 2025), formal version of Lemma 5.8)**.** *If we have the below conditions,*

- **Condition 1.** *Let $\mathcal{D}$ and $\mathcal{D}'$ are neighboring dataset (see Definition 3.6).*

- **Condition 2.** *Let $H := A^\top W^{1-2/p} A$ denotes the symmetric positive semi-definite matrix generated by $X$, and $H'$ denotes the symmetric positive definite matrix generated by neighboring dataset $X'$.*

- **Condition 3.** *Let $\epsilon_\alpha \in (0,1)$ and $\delta_\alpha \in (0,1)$ denote the DP parameter for $A^\top W^{1-2/p} A$.*

- **Condition 4.** *Let $\mathcal{Y}, \mathcal{Y}'$ denote neighboring datasets, which differ by a single data element.*

- **Condition 5.** *Let $\Delta$ be defined in Definition 5.2 and $\Delta < 1$.*

- **Condition 6.** *Let $M, \mathcal{M}$ be defined in Definition 5.1 and $M \leq \Delta$.*

- **Condition 7.** *Let the input $\Sigma = \mathcal{M}(\mathcal{Y})$.*

- **Condition 8.** *Let $\rho = O(\sqrt{(n^2 + \log(1/\gamma))/k} + (n^2 + \log(1/\gamma))/k)$.*

- **Condition 9.** *Let $k \in \mathbb{N}$.*

- **Condition 10.** *Let $\gamma \in (0,1)$.*

- **Condition 11.** *Let $\eta_{\max} I_{n \times n} \succeq H \succeq \eta_{\min} I_{n \times n}$, for some $\eta_{\max}, \eta_{\min} \in \mathbb{R}$.*

- **Condition 12.** *Let $\widetilde{H}$ denote the private $H$ generated by Algorithm 4 with $H$ as the input.*

- **Condition 13.** *Let $\sqrt{n}\psi/\eta_{\min} < \Delta$, where $\Delta$ is defined in Definition 5.2.*

*Then, there exists an Algorithm 4 such that*

- *Part 1. Algorithm 4 is $(\epsilon_\alpha, \delta_\alpha)$-DP.*

- *Part 2. Outputs $\widehat{\Sigma} \in \mathbb{S}_+^n$ such that with probabilities at least $1 - \gamma$,*

$$\|\Sigma^{-1/2} \widehat{\Sigma} \Sigma^{-1/2} - I_n\|_F \leq \rho$$

- *Part 3.*

$$(1 - \rho)\Sigma \preceq \widehat{\Sigma} \preceq (1 + \rho)\Sigma.$$

## C  PROOF OF MAIN RESULT

**Lemma C.1** (Correctness of the algorithm, formal version of Lemma 4.3)**.** *The output of* UPDATE$(w)$ *in Algorithm 2 satisfies*

$$M = A^\top (AV^{1-2/p}A^\top)^{-1} A \text{ and}$$
$$Q = MV^{1/2-1/p}R^\top$$

*The output of* QUERY$(h)$ *in Algorithm 3 satisfies*

$$p_s = \widetilde{P}(R^\top)_{*,l} R_{l,*} h$$
$$p_x = (I - \widetilde{P})(R^\top)_{*,l} R_{l,*} h,$$

*where $\widetilde{P} = V^{1/2-1/p} A^\top (A\widetilde{V}^{1-2/p}A^\top)^{-1} A\widetilde{V}^{1/2-1/p}$, and $\widetilde{V}$ is outputted by* UPDATE$(w)$.

*Proof.* Let $S$ denote the support of $\Delta$.

Thus, by the Woodbury matrix identity (Fact 3.1) and definition of $M^{\mathrm{new}}$, we have

$$A^\top (A(V^{\mathrm{new}})^{1-2/p}A^\top)^{-1} A$$
$$= A^\top (A(V + \Delta)^{1-2/p}A^\top)^{-1} A$$
$$= A^\top ((AV^{1-2/p}A^\top)^{-1} - (AV^{1-2/p}A^\top)^{-1} A_{*,S} \cdot (\Delta_{S,S}^{-1} + (A^\top)_{S,*}(AV^{1-2/p}A^\top)^{-1} A_{*,S})^{-1}$$

$$\cdot (A^\top)_{S,*}(AV^{1-2/p}A^\top)^{-1})A$$
$$= A^\top(AV^{1-2/p}A^\top)^{-1}A - A^\top(AV^{1-2/p}A^\top)^{-1}A_{*,S}$$
$$\cdot (\Delta_{S,S}^{-1} + (A^\top)_{S,*}(AV^{1-2/p}A^\top)^{-1}A_{*,S})^{-1} \cdot (A^\top)_{S,*}(AV^{1-2/p}A^\top)^{-1}A$$
$$= M - M_{*,S}(\Delta_{S,S}^{-1} + M_{S,S})^{-1}M_{S,*}$$
$$= M^{\text{new}}.$$

where the first step follows from the definition of $V^{\text{new}} = V + \Delta$, the second step follows from the Fact 3.1, the third step distributes the terms, the fourth step follows from the definition of $M = A^\top(AV^{1-2/p}A^\top)^{-1}A$, and the last step follows from the definition of $M^{\text{new}}$.

Note the output $M = M^{\text{new}}$ and $V = V^{\text{new}}$, so we have the output satisfying $M = A^\top(AVA^\top)^{-1}A$.

As for $Q$, notice by definition

$$Q^{\text{new}} = Q + (M^{\text{new}} \cdot \Gamma) \cdot R^\top + (M^{\text{new}} - M) \cdot V^{1/2-1/p} \cdot R^\top$$
$$= MV^{1/2-1/p}R^\top + (M^{\text{new}} \cdot \Gamma) \cdot R^\top + (M^{\text{new}} - M) \cdot V^{1/2-1/p} \cdot R^\top$$
$$= MV^{1/2-1/p}R^\top + M^{\text{new}}((V^{\text{new}})^{1/2-1/p} - V^{1/2-1/p})R^\top + (M^{\text{new}} - M)V^{1/2-1/p}R^\top$$
$$= M^{\text{new}}((V^{\text{new}})^{1/2-1/p} - V^{1/2-1/p})R^\top + M^{\text{new}}V^{1/2-1/p}R^\top$$
$$= M^{\text{new}}(V^{\text{new}})^{1/2-1/p}R^\top$$

where the first step follows from the definition of $Q^{\text{new}}$, the second step follows from definition of $Q$, the third step follows from the definition of $\Gamma$, the fourth step distributes and eliminates the term $MV^{1/2-1/p}R^\top$, and the last step distributes and eliminates the term $M^{\text{new}}V^{1/2-1/p}R^\top$.

Again, since the output $Q = Q^{\text{new}}$, $M = M^{\text{new}}$ and $V = V^{\text{new}}$, we have the output satisfying $Q = MV^{1/2-1/p}R^\top$.

For $\text{QUERY}(h)$ procedure, by definition we have

$$p_m = \widetilde{V}^{1/2-1/p} \cdot (M_{*,\widetilde{S}}) \cdot (\widetilde{\Delta}_{\widetilde{S},\widetilde{S}}^{-1} + M_{\widetilde{S},\widetilde{S}})^{-1} \cdot (Q_{\widetilde{S},l} + M_{\widetilde{S},*} \cdot \widetilde{\Gamma} \cdot (R^\top)_{*,l}) \cdot R_{l,*} \cdot h$$
$$= \widetilde{V}^{1/2-1/p} \cdot (M_{*,\widetilde{S}}) \cdot (\widetilde{\Delta}_{\widetilde{S},\widetilde{S}}^{-1} + M_{\widetilde{S},\widetilde{S}})^{-1}$$
$$\cdot ((MV^{1/2-1/p}R^\top)_{\widetilde{S},l} + M_{\widetilde{S},*} \cdot (\widetilde{V}^{1/2-1/p} - V^{1/2-1/p}) \cdot (R^\top)_{*,l}) \cdot R_{l,*} \cdot h$$
$$= \widetilde{V}^{1/2-1/p} \cdot (M_{*,\widetilde{S}}) \cdot (\widetilde{\Delta}_{\widetilde{S},\widetilde{S}}^{-1} + M_{\widetilde{S},\widetilde{S}})^{-1} \cdot M_{\widetilde{S},*} \cdot \widetilde{V}^{1/2-1/p} \cdot (R^\top)_{*,l} \cdot h, \tag{3}$$

where the first step follows from the definition of $p_m$, the second step follows from the definition of $Q$ and $\widetilde{\Gamma}$, and the third step eliminates the terms.

Thus,

$$p_s = \widetilde{V}^{1/2-1/p} \cdot (Q_{*,l} + M \cdot \widetilde{\Gamma} \cdot (R^\top)_{*,l}) \cdot R_{l,*} \cdot h - p_m$$
$$= \widetilde{V}^{1/2-1/p} \cdot ((MV^{1/2-1/p}R^\top)_{*,l} + M \cdot (\widetilde{V}^{1/2-1/p} - V^{1/2-1/p}) \cdot (R^\top)_{*,l}) \cdot R_{l,*} \cdot h - p_m$$
$$= \widetilde{V}^{1/2-1/p} \cdot M \cdot \widetilde{V}^{1/2-1/p} \cdot (R^\top)_{*,l} \cdot R_{l,*} \cdot h - p_m$$
$$= \widetilde{V}^{1/2-1/p}(M - M_{*,\widetilde{S}}(\widetilde{\Delta}_{\widetilde{S},\widetilde{S}}^{-1} + M_{\widetilde{S},\widetilde{S}})^{-1}M_{\widetilde{S},*})\widetilde{V}^{1/2-1/p}(R^\top)_{*,l}R_{l,*}h, \tag{4}$$

where the first step follows the definition of $p_s$, the second step follows from the definition of $Q$ and $\widetilde{\Gamma}$, the third step follows from eliminates the terms, and the last step substitutes $p_m$ by Eq. (3).

Note $\widetilde{V}$ only differs from $V$ in entries corresponding to the set $\widetilde{S}$, again by the Woodbury matrix identity (Fact 3.1) and the definition of $M$, we have

$$A^\top(A\widetilde{V}^{1-2/p}A^\top)^{-1}A$$
$$= A^\top(A(V^{1-2/p} + \widetilde{\Delta})A^\top)^{-1}A$$
$$= A^\top((AV^{1-2/p}A^\top)^{-1} - (AV^{1-2/p}A^\top)^{-1}A_{*,\widetilde{S}}$$

$$\cdot (\widetilde{\Delta}_{\widetilde{S},\widetilde{S}}^{-1} + (A^\top)_{\widetilde{S},*}(AV^{1-2/p}A^\top)^{-1}A_{*,\widetilde{S}})^{-1} \cdot (A^\top)_{\widetilde{S},*}(AV^{1-2/p}A^\top)^{-1})A$$

$$= A^\top(AV^{1-2/p}A^\top)^{-1}A - A^\top(AV^{1-2/p}A^\top)^{-1}A_{*,\widetilde{S}}$$

$$\cdot (\widetilde{\Delta}_{\widetilde{S},\widetilde{S}}^{-1} + (A^\top)_{\widetilde{S},*}(AV^{1-2/p}A^\top)^{-1}A_{*,\widetilde{S}})^{-1} \cdot (A^\top)_{\widetilde{S},*}(AV^{1-2/p}A^\top)^{-1}A$$

$$= M - M_{*,\widetilde{S}}(\widetilde{\Delta}_{\widetilde{S},\widetilde{S}}^{-1} + M_{\widetilde{S},\widetilde{S}})^{-1}M_{\widetilde{S},*}, \tag{5}$$

where the first step follows from $\widetilde{V}^{1-2/p} = V^{1-2/p} + \widetilde{\Delta}$, the second step follows from Fact 3.1 , the third step distributes two terms, and the last step follows from the definition of $M$, which implies

$$p_s = \widetilde{V}^{1/2-1/p}A^\top(A\widetilde{V}^{1-2/p}A^\top)^{-1}A\widetilde{V}^{1/2-1/p}(R^\top)_{*,l}R_{l,*}h$$

$$= \widetilde{P}(R^\top)_{*,l}R_{l,*}h, \tag{6}$$

where the first step follows from Eq. (4) and (5), and the second step follows from the definition of $\widetilde{P}$.

Further,

$$p_x = (R^\top)_{*,l}R_{l,*}h - p_s$$

$$= (I - \widetilde{P})(R^\top)_{*,l}R_{l,*}h,$$

where the first step follows from the definition of $p_x$, and the second step follows from Eq. (6).

Thus we completes the proof. $\qquad\square$

**Lemma C.2** (Query time, formal version of Lemma 4.6)**.** *The query time of data-structure* MAIN-TAINPROJECTION *(Algorithm 1) is $O(n^{1+b+o(1)} + n^{1+a+o(1)})$.*

*Proof.* Notice by the algorithm we have $|\widetilde{S}| \leq n^a$. Thus, $\widetilde{\Gamma}$ is a sparse diagonal matrix with at most $n^a$ non-zero elements. The running time mainly comes from three parts.

**Part 1.** Computing $p_m$:

- Compute $R_{l,*} \cdot h$: matrix-vector multiplication between matrix of size $n^b \times n$ and vector of size $n \times 1$, this takes $n^{1+b}$ time.

- Compute $(R^\top)_{*,l} \cdot (R_{l,*}h)$: matrix-vector multiplication between matrix of size $n \times n^b$ and vector of size $n^b \times 1$, this takes $n^{1+b}$ time.

- Compute $\widetilde{\Gamma} \cdot (R_{l,*}^\top R_{l,*}h)$: matrix-vector multiplication between sparse diagonal matrix with at most $n^a$ non-zero elements and vector of size $n \times 1$, this takes $n^a$ time.

- Compute $M_{\widetilde{S},*} \cdot (\widetilde{\Gamma}R_{l,*}^\top h)$: matrix-vector multiplication between matrix of size at most $n^a \times n$ and sparse vector with at most $n^a$ non-zero elements, this takes $n^{2a}$ time.

- Compute $Q_{\widetilde{S},l} \cdot (R_{l,*}h)$: matrix-vector multiplication between matrix of size at most $n^a \times n^b$ and vector of size $n^b \times 1$, this takes $n^{a+b}$ time.

- Compute $(\widetilde{\Delta}_{\widetilde{S},\widetilde{S}}^{-1} + M_{\widetilde{S},\widetilde{S}})^{-1}$: inverse of matrix of size at most $n^a \times n^a$, this takes $n^{a\omega}$ time.

- Compute $(\widetilde{\Delta}_{\widetilde{S},\widetilde{S}}^{-1} + M_{\widetilde{S},\widetilde{S}})^{-1} \cdot [(Q_{\widetilde{S},l} + M_{\widetilde{S},*}\widetilde{\Gamma}(R^\top)_{*,l})R_{l,*}h]$: matrix-vector multiplication between matrix of size at most $n^a \times n^a$ and vector of size at most $n^a \times 1$, this takes $n^{2a}$ time.

- Compute $\widetilde{V}^{1/2-1/p} \cdot (M_{*,\widetilde{S}})$: matrix-matrix multiplication between diagonal matrix of size $n \times n$ and matrix of size at most $n \times n^a$, this takes $n^{1+a}$ time.

- Compute $[\widetilde{V}^{1/2-1/p}M_{*,\widetilde{S}}] \cdot [(\widetilde{\Delta}_{\widetilde{S},\widetilde{S}}^{-1} + M_{\widetilde{S},\widetilde{S}})^{-1}(Q_{\widetilde{S},l} + M_{\widetilde{S},*}\widetilde{\Gamma}(R^\top)_{*,l})R_{l,*}h]$: matrix-vector multiplication between matrix of size at most $n \times n^a$ and vector of size at most $n^a \times 1$, this takes $n^{1+a}$ time.

To conclude, we can compute $p_m$ in $O(n^{1+b} + n^{a\omega} + n^{1+a})$ time.

**Part 2.** Computing $p_s$:

- Compute $R_{l,*}h$ and $\widetilde{\Gamma} R_{l,*}^\top R_{l,*} h$ in same way as in calculating $p_m$: take $n^{1+b}$ and $O(n^{1+b} + n^a)$ time respectively.

- Compute $\widetilde{V}^{1/2-1/p} \cdot Q_{*,l}$: matrix-matrix multiplication between diagonal matrix of size $n \times n$ and matrix of size $n \times n^b$, takes $n^{1+b}$ time.

- Compute $[\widetilde{V}^{1/2-1/p} Q_{*,l}] \cdot [R_{l,*}h]$: matrix-vector multiplication between matrix of size $n \times n^b$ and vector of size $n^b \times 1$, takes $n^{1+b}$ time.

- Compute $M \cdot [\widetilde{\Gamma} R_{l,*}^\top R_{l,*} h]$: matrix-vector multiplication between matrix of size $n \times n$ and sparse vector with at most $n^a$ non-zero elements, takes $O(n^{1+a})$ time.

- Compute $\widetilde{V}^{1/2-1/p} \cdot [M\widetilde{\Gamma} R_{l,*}^\top R_{l,*} h]$: matrix-vector multiplication between diagonal matrix of size $n \times n$ and vector of size $n \times 1$, takes $n$ time.

To conclude, we can compute $p_s$ in $O(n^{1+b} + n^{1+a})$ time.

**Part 3.** Computing $p_x$:

- Compute $R_{l,*}^\top R_{l,*} h$ in same way as in calculating $p_m$: take $O(n^{1+b})$ time.

Thus, the overall running time is

$$O(n^{1+a} + n^{1+b} + n^{a\omega}).$$

Finally, we note that $\omega \le 3 - \alpha \le 3 - a$ (see (Cohen et al., 2021b)) and hence $a \cdot \omega \le a(3 - a) \le (1 + a)$. Therefore, the final running time it takes is $O(b^{1+b+o(1)} + n^{1+a+o(1)})$. $\qquad\square$

# D   PROOF OF DIFFERENTIAL PRIVACY GUARANTEES

**Lemma D.1** (Sensitivity of $W^{1/2-1/p}A$, formal version of Lemma 5.3). *If the following conditions hold:*

- *Let the neighboring dataset $X$ and $X'$ be defined in Definition 3.6.*

- *Let $\epsilon_J, \delta_J \in \mathbb{R}$ denote the DP parameters for $J$.*

- *Let $J := W^{1/2-1/p}A$ denote the data generated by $X$ and $J'$ denote the data generated by neighboring dataset $X'$, where $W^{1/2-1/p} \in \mathbb{R}^{m \times m}$ and $A \in \mathbb{R}^{m \times n}$.*

- *Let $\beta > 0$ be defined as Definition 3.6.*

*Then, we can show that the sensitivity of $J$ is $\sqrt{n} \cdot \beta$.*

*Proof.* Without loss of generality, we use $x_m \in \mathbb{R}^n$ and $x'_m \in \mathbb{R}^n$ to denote the different items in $X$ and $X'$. According to the definition of the neighboring dataset, we have

$$\|x_m - x'_m\|_2 \le \beta.$$

Then, we have

$$\begin{aligned}
\|J - J'\|_1 &= \|x_m - x'_m\|_1 \\
&\le \sqrt{n} \cdot \|x_m - x'_m\|_2 \\
&= \sqrt{n} \cdot \beta,
\end{aligned}$$

where the first step follows from $\|u - v\|_1 \le \sqrt{n}\|u - v\|_2$ for any $u, v \in \mathbb{R}^n$, and the second step follows from $\|x_m - x'_m\|_2 \le \beta$. $\qquad\square$

**Lemma D.2** (DP guarantees for $W^{1/2-1/p}A(A^\top W^{1-2/p}A)^{-1}A^\top W^{1/2-1/p} \cdot h$, formal version of Lemma 5.10)**.** *If the following conditions hold:*

- *Let $\epsilon_\alpha, \delta_\alpha \in \mathbb{R}$ denote the DP parameter for $A^\top W^{1-2/p}A$.*

- *Let $\epsilon_J, \delta_J \in \mathbb{R}$ denote the DP parameters for $W^{1/2-1/p}A$ and $A^\top W^{1/2-1/p}$.*

- *Let $\epsilon = 2\epsilon_J + \epsilon_\alpha$.*

- *Let $\delta = 2\delta_J + \delta_\alpha$.*

- *Let $H$ and $\widetilde{H}$ be defined as Lemma 5.9.*

- *Let $J$ and $\widetilde{J}$ be defined as Lemma 5.4.*

*Then, we can show $W^{1/2-1/p}A(A^\top W^{1-2/p}A)^{-1}A^\top W^{1/2-1/p} \cdot h$ is $(\epsilon, \delta)$-DP.*

*Proof.* Since we have

- $A^\top W^{1-2/p}A$ is $(\epsilon_\alpha, \delta_\alpha)$-DP.

- $W^{1/2-1/p}A$ is $(\epsilon_J, \delta_J)$-DP.

- $A^\top W^{1/2-1/p}$ is $(\epsilon_J, \delta_J)$-DP.

- $\epsilon = 2\epsilon_J + \epsilon_\alpha, \delta = 2\delta_J + \delta_\alpha$.

By Lemma 3.9, we have $W^{1/2-1/p}A(A^\top W^{1-2/p}A)^{-1}A^\top W^{1/2-1/p}$ is $(\epsilon, \delta)$-DP.

Next, by the post-processing property of differential privacy (Lemma 3.7), we conclude that $W^{1/2-1/p}A(A^\top W^{1-2/p}A)^{-1}A^\top W^{1/2-1/p} \cdot h$ is also $(\epsilon, \delta)$-DP.

Thus, we complete the proof. $\qquad\square$

# E PROOF OF UTILITY GUARANTEES

**Lemma E.1** (Utility guarantees for $W^{1/2-1/p}A$, formal version of 5.5)**.** *If the following conditions hold:*

- *Let the neighboring dataset $X$ and $X'$ be defined in Definition 3.6.*

- *Let $\epsilon_J, \delta_J \in \mathbb{R}$ denote the DP parameters for $J$.*

- *Let $\Delta_J := \sqrt{n} \cdot \beta$ denote the sensitivity of $J$.*

- *Let $J := W^{1/2-1/p}A$ denote the data generated by $X$, where $W^{1/2-1/p} \in \mathbb{R}^{m \times m}$ and $A \in \mathbb{R}^{m \times n}$.*

- *Let $B_L = (\Delta_J/\epsilon_J)\log(1 + \frac{\exp(\epsilon_J)-1}{2\delta_J})$.*

- *Let $\widetilde{J} := J + \mathrm{TLap}(\Delta_J, \epsilon_J, \delta_J)$.*

*Then, we can show that*

$$\|\widetilde{J} - J\|_2 \le \sqrt{n} \cdot B_L.$$

*Proof.* For $i \in [m], j \in [n]$, let $J(i,j), J'(i,j) \in \mathbb{R}$ denote the $(i,j)$-th entry of $J$ and $J'$, respectively. Let $J_i \in \mathbb{R}^n$ denotes the $i$-th column of $J$.

By the definition of $\widetilde{J}$, we have

$$\widetilde{J}(i,j) = J(i,j) + \mathrm{TLAP}(\Delta_J, \epsilon_J, \delta_J)$$

Recall that we have $B_L = (\Delta_J/\epsilon_J)\log(1+\frac{e^{\epsilon_J}-1}{2\delta_J})$. By the definition of truncated Laplace, we have

$$|\mathrm{TLAP}(\Delta_J,\epsilon_J,\delta_J)| \le B_L.$$

Combining the above two equations, for $i \in [m]$, we have

$$\|\widetilde{J} - J\|_2 \le \sqrt{n} \cdot B_L.$$

Thus, we complete the proof. $\square$

**Lemma E.2** (Utility guarantees for $A^\top W^{1/2-1/p}$, formal version of Lemma 5.7)**.** *If the following conditions hold:*

- *Let the neighboring dataset $X$ and $X'$ be defined in Definition 3.6.*

- *Let $\epsilon_J, \delta_J \in \mathbb{R}$ denote the DP parameters for $J$.*

- *Let $\Delta_J := \sqrt{n} \cdot \beta$ denote the sensitivity of $J$.*

- *Let $J^\top := A^\top W^{1/2-1/p}$ denote the data generated by $X$, where $W^{1/2-1/p} \in \mathbb{R}^{m \times m}$ and $A^\top \in \mathbb{R}^{n \times m}$.*

- *Let $\widetilde{J}^\top := J^\top + \mathrm{TLap}(\Delta_J,\epsilon_J,\delta_J)$.*

- *Let $B_L = (\Delta_J/\epsilon_J)\log(1+\frac{\exp(\epsilon_J)-1}{2\delta_J})$.*

*we can show that*

$$\|\widetilde{J}^\top - J^\top\|_2 \le \sqrt{n} \cdot B_L.$$

*Proof.* From Lemma 5.5, we have

$$\|\widetilde{J} - J\|_2 \le \sqrt{n} \cdot B_L.$$

Then,

$$\|\widetilde{J}^\top - J^\top\|_2 = \|(\widetilde{J} - J)^\top\|_2$$
$$= \|\widetilde{J} - J\|_2$$
$$\le \sqrt{n} \cdot B_L.$$

Where the first step follows from the invariance of the norm under transposition, the second step follows from the norm property $\|A^\top\|_2 = \|A\|_2$, and the third step follows from Lemma 5.5.

Thus, we complete the proof. $\square$

**Lemma E.3** (Utility guarantees for $(A^\top W^{1-2/p}A)^{-1}$, formal version of Lemma 5.9)**.** *If the following conditions hold:*

- **Condition 1.** *If $\mathcal{D} \in \mathbb{R}^{n \times d}$ and $\mathcal{D}' \in \mathbb{R}^{n \times d}$ are neighboring dataset (see Definition 3.6)*

- **Condition 2.** *Let $H := A^\top W^{1-2/p}A$ denotes the symmetric positive semi-definite matrix generated by $\mathcal{D}$.*

- **Condition 3.** *Let $\widetilde{H}$ denote the private $H$ generated by Algorithm 4 with $H$ as the input.*

- **Condition 4.** *Let $\eta_{\max}I_{n \times n} \succeq H \succeq \eta_{\min}I_{n \times n}$, for some $\eta_{\max}, \eta_{\min} \in \mathbb{R}$.*

- **Condition 5.** *Let $\sqrt{n}\psi/\eta_{\min} < \Delta$, where $\Delta$ is defined in Definition 5.2.*

- **Condition 6.** *Let $\rho = O(\sqrt{(n^2 + \log(1/\gamma))/k} + (n^2 + \log(1/\gamma))/k)$.*

- **Condition 7.** *Let $\gamma \in (0,1)$.*

*Then, with probability $1 - \gamma$, we have*

$$\|H^{-1} - \widetilde{H}^{-1}\| \le O(\frac{\rho \cdot \eta_{\max}}{\eta_{\min}^2})$$

*Proof.* We consider the $\|H^{-1}\|$ term. We have

$$\|H^{-1}\| = \|(A^\top W^{1-2/p} A)^{-1}\| \tag{7}$$

$$= \sigma_{\max}((A^\top W^{1-2/p} A)^{-1})$$

$$= \frac{1}{\sigma_{\min}((A^\top W^{1-2/p} A))}$$

$$\leq \frac{1}{\eta_{\min}} \tag{8}$$

where the 1st step is due to Condition 2, the 2nd step is because of definition of spectral norm, the 3rd step is due to $\sigma_{\max}(A^{-1}) = 1/\sigma_{\min}(A)$ holds for any matrix $A$, the 4th step is from $K \succeq \eta_{\min} I_{n \times n}$.

Similarly, we can have

$$\|\widetilde{H}^{-1}\| \leq \frac{1}{\eta_{\min}} \tag{9}$$

Recall in Lemma A.3, we have

$$\|H - \widetilde{H}\| \leq \rho \cdot \eta_{\max} \tag{10}$$

Then, by Lemma A.4, we have

$$\|H^{-1} - \widetilde{H}^{-1}\| \leq O(\max\{\|H^{-1}\|^2, \|\widetilde{H}^{-1}\|^2\} \cdot \|H - \widetilde{H}\|)$$

$$\leq O(\frac{1}{\eta_{\min}^2} \cdot \|H - \widetilde{H}\|)$$

$$\leq O(\frac{\rho \cdot \eta_{\max}}{\eta_{\min}^2})$$

where the 1st step is because of Lemma A.4, the 2nd step is due to Eq. (7) and Eq. (9), the 3rd step is from Eq. (10). $\qquad\square$

**Lemma E.4** (Utility guarantees for $W^{1/2-1/p} A(A^\top W^{1-2/p} A)^{-1} A^\top W^{1/2-1/p} \cdot h$, formal version of Lemma 5.11). *If the following conditions hold:*

- *If $\mathcal{D} \in \mathbb{R}^{n \times d}$ and $\mathcal{D}' \in \mathbb{R}^{n \times d}$ are neighboring dataset (see Definition 3.6)*

- *Let $H$ and $\widetilde{H}$ be defined as Lemma 5.9.*

- *Let $J$ and $\widetilde{J}$ be defined as Lemma 5.4.*

- *Let $\sigma_J := \|J\|_2$ denotes the $\ell_2$ norm of $J$.*

- *Let $\sigma_h := \|h\|_2$ denotes the $\ell_2$ norm of $h$.*

- *Let $\sigma_{H^{-1}} := \|H^{-1}\|_2$ denotes the $\ell_2$ norm of $H^{-1}$.*

- *Let $\eta_{\max} I_{n \times n} \succeq H \succeq \eta_{\min} I_{n \times n}$, for some $\eta_{\max}, \eta_{\min} \in \mathbb{R}$.*

- *Let $\sqrt{n}\psi/\eta_{\min} < \Delta$, where $\Delta$ is defined in Definition 5.2.*

- *Let $\rho = O(\sqrt{(n^2 + \log(1/\gamma))/k} + (n^2 + \log(1/\gamma))/k)$.*

- *Let $\gamma \in (0, 1)$.*

- *Let $B_L \in \mathbb{R}$ be defined in Lemma 5.5.*

*Then, with probability $1 - \gamma$, we have*

$$|JH^{-1}J^\top \cdot h - \widetilde{J}\widetilde{H}^{-1}\widetilde{J}^\top \cdot h| \leq 2\sigma_J \sigma_h \sqrt{n} \cdot B_L + \sigma_J^2 \sigma_h \cdot O(\frac{\rho \cdot \eta_{\max}}{\eta_{\min}^2}).$$

*Proof.* We have

$$|JH^{-1}J^\top \cdot h - \widetilde{J}\widetilde{H}^{-1}\widetilde{J}^\top \cdot h|$$

$$= |(JH^{-1}J^\top - \widetilde{J}\widetilde{H}^{-1}\widetilde{J}^\top) \cdot h|$$

$$= |((JH^{-1}J^\top - \widetilde{J}H^{-1}J^\top) + (\widetilde{J}H^{-1}J^\top - \widetilde{J}\widetilde{H}^{-1}J^\top) + (\widetilde{J}\widetilde{H}^{-1}J^\top - \widetilde{J}\widetilde{H}^{-1}\widetilde{J}^\top)) \cdot h|$$

$$\leq (|JH^{-1}J^\top - \widetilde{J}H^{-1}J^\top| + |\widetilde{J}H^{-1}J^\top - \widetilde{J}\widetilde{H}^{-1}J^\top| + |\widetilde{J}\widetilde{H}^{-1}J^\top - \widetilde{J}\widetilde{H}^{-1}\widetilde{J}^\top|) \cdot \|h\|_2, \quad (11)$$

where the first step follows from basic algebra, the second step follows from basic algebra, the third step follows from triangle inequality.

Consider $|JH^{-1}J^\top - \widetilde{J}H^{-1}J^\top|$, we have:

$$|JH^{-1}J^\top - \widetilde{J}H^{-1}J^\top|$$

$$\leq \|J - \widetilde{J}\|_2\|H^{-1}\|_2\|J^\top\|_2$$

$$\leq \sigma_{H^{-1}}\sigma_J\sqrt{n} \cdot B_L, \quad (12)$$

where the first step follows from Cauchy–Schwarz inequality, the second step follows from Lemma 5.5.

Consider $|\widetilde{J}H^{-1}J^\top - \widetilde{J}\widetilde{H}^{-1}J^\top|$, we have:

$$|\widetilde{J}H^{-1}J^\top - \widetilde{J}\widetilde{H}^{-1}J^\top|$$

$$\leq \|\widetilde{J}\|_2|H^{-1} - \widetilde{H}^{-1}|\|J^\top\|_2$$

$$\leq \sigma_J^2 \cdot O(\frac{\rho \cdot \eta_{\max}}{\eta_{\min}^2}), \quad (13)$$

where the first step follows from Cauchy–Schwarz inequality, the second step follows from Lemma 5.9.

Consider $|\widetilde{J}\widetilde{H}^{-1}J^\top - \widetilde{J}\widetilde{H}^{-1}\widetilde{J}^\top|$, we have:

$$|\widetilde{J}\widetilde{H}^{-1}J^\top - \widetilde{J}\widetilde{H}^{-1}\widetilde{J}^\top|$$

$$\leq \|\widetilde{J}\|_2\|\widetilde{H}^{-1}\|_2\|J^\top - \widetilde{J}^\top\|_2$$

$$\leq \sigma_{H^{-1}}\sigma_J\sqrt{n} \cdot B_L, \quad (14)$$

where the first step follows from Cauchy–Schwarz inequality, the second step follows from Lemma 5.7.

Combine the equations above, we have:

$$|JH^{-1}J^\top \cdot h - \widetilde{J}\widetilde{H}^{-1}\widetilde{J}^\top \cdot h|$$

$$\leq (|JH^{-1}J^\top - \widetilde{J}H^{-1}J^\top| + |\widetilde{J}H^{-1}J^\top - \widetilde{J}\widetilde{H}^{-1}J^\top| + |\widetilde{J}\widetilde{H}^{-1}J^\top - \widetilde{J}\widetilde{H}^{-1}\widetilde{J}^\top|) \cdot \|h\|_2$$

$$\leq (\sigma_{H^{-1}}\sigma_J\sqrt{n} \cdot B_L + \sigma_J^2 \cdot O(\frac{\rho \cdot \eta_{\max}}{\eta_{\min}^2}) + \sigma_{H^{-1}}\sigma_J\sqrt{n} \cdot B_L) \cdot \sigma_h$$

$$= 2\sigma_J\sigma_h\sqrt{n} \cdot B_L + \sigma_J^2\sigma_h \cdot O(\frac{\rho \cdot \eta_{\max}}{\eta_{\min}^2}),$$

where the first step follows from Eq (11), the second step follows from Eq (12), Eq (13) and Eq (14), the third step follows from basic algebra.

Thus, we complete the proof. $\square$

## LLM USAGE DISCLOSURE

LLMs were used only to polish language, such as grammar and wording. These models did not contribute to idea creation or writing, and the authors take full responsibility for this paper's content.

