# OpenReview forum: "Dynamic Weighted Projection Maintenance with $\ell^p$-Lewis Weight"
_ICLR.cc/2026/Conference — Submitted to ICLR 2026_

### Official Review · Reviewer_A9Pp · 2025-10-23

**Soundness:** 2
**Presentation:** 1
**Contribution:** 1
**Rating:** 2
**Confidence:** 3

**Summary:**

The paper studies a specific instantiation of Dynamic Projection Maintenance, which in its most general form concerns maintaining a projection $P(B) = B(B^T B)^{-1}B^T$ under low-rank updates to $B$ and supporting queries of the form $P(B)h$ for arbitrary online vectors $h$. In linear programming, the setting where $B=\sqrt{\mathrm{diag}(w)}A$ for $w$ a vector encoding slack variables, $A$ encoding the constraint matrix and where the updates are made to $w$, projection maintenance is a key subroutine. This paper studies a generalized version of this problem, where instead $B=\mathrm{diag}(w)^{1/2-1/p} A$ for $p>0$, for which they give an algorithm and argue for how it can be implemented to satisfy differential privacy.

**Strengths:**

Dynamic Projection Maintenance is an interesting problem, and I think it can be argued to be within scope for ICLR.

**Weaknesses:**

1. The exposition of the paper needs more work. Most of the main paper constitutes chains of lemmas/algorithms with limited interpretation or intuition. After reading the paper from start to finish I find it hard to say how the contribution was achieved, what exactly it consisted of, and how it compares to past work. On a related note, the main problems being investigated, Definition 1.2, seem to be defined for general sparse/low-rank updates, but the main algorithm seems to only consider the case where the updates are diagonal matrices.
2. While I agree that the proposed problem is a generalization, the paper provides no intuition for what the challenge in this extended setting is. *Naively*, given that the only place where $p$ influences the problem is to what power diagonal matrices are taken to (again, Definition 1.2 seems to support non-diagonal $W$, but here I am referencing the setting in which the algorithm seems to operate), I would expect that the generalized problem is not much harder unless argued otherwise.
2. The writing needs more work, and appears excessively inspired from past work in places. Fact 3.1 stating the Woodbury matrix identity also includes a statement about $M_{w^{new}}$ which has not yet been defined, and seemingly is never used. Looking up the arXiv version of (Cohen et al., 2021), arXiv:1810.07896, their Fact 2.1 states the Woodbury matrix identity, with the text immediately after matching the content of Fact 3.1 here, including their eq. (9) introducing $M_{w^{new}}$. Lemma 4.5 for the update time, which cites (Cohen et al., 2021) for the proof and which indeed appears to correspond to Lemma 5.4 in the arXiv version, contains the term $g_r$, which is not commented on or defined. $g_r$ comes from a potential argument in the arXiv version, eq. (20). The main result Theorem 4.1 appears like Theorem 5.1 from the arXiv version of (Cohen et al., 2021) transcribed to this setting and techniques, including the same footnote and typo “with the expectation and variance is [...]”. I do not find these issues too severe, but it is sloppy and makes the work appear derivative even if it were to be meaningfully novel.
3. The claimed DP guarantee is not clearly expressed for the problem. From what I can tell, the algorithm provided (split up into Algorithm 1-3) does not implement DP. My impression is that the authors argue that their algorithm can be implemented with DP by amending the algorithm, but this does not appear clearly expressed.

**Questions:**

Given that the contribution and its novelty is hard to assess in the current write-up, my initial recommendation is rejection. The presentation of the paper needs more work.

I ask the following questions to better understand the character and quality of the contribution in the paper.
1. What is the challenge for this more general setting over the case where $B=\sqrt{\mathrm{diag}(w)}A$, corresponding to $p\to\infty$?
2. What new techniques do you need to employ compared to (Cohen et al., 2021)?
3. How should Theorem 4.1 be interpreted with respect to Theorem 5.1 from (Cohen et al., 2021, arXiv:1810.07896)?
4. Has Dynamic Projection Maintenance (and in particular the version from Cohen et al., 2021) been studied under differential privacy before?

Typos:
- (Cohen et al., 2021) and (Song et al., “Sketching meets DP”, 2023) appear twice in the bibliography.
- Unclear citations in related work e.g., “Mahabadi et al.(Makarychev et al., 2022)” and “Chen et al.(Cohen et al., 2019)”.

---

### Official Review · Reviewer_hiEX · 2025-10-29

**Soundness:** 2
**Presentation:** 2
**Contribution:** 2
**Rating:** 2
**Confidence:** 4

**Summary:**

The paper studies the general problem of maintaining the Lewis Weight Projection matrix, i.e., $W^{1/2−1/p}A(A^T W^{1-2/p}A)^{-1}A^T W^{1/2-1/p}$ as $W$ undergoes updates. This generalizes the case of $p=\infty$ which was used in Cohen et al’19 for linear programming. Their algorithm is deterministic and only requires sublinear update time. They further prove privacy guarantees for the algorithm. Their algorithm uses sketching matrices and their update step mainly maintains the sketched version of the matrices.

Overall, while the algorithm is simple, I have certain concerns about the problem and the analysis (see Questions below). My scores are due to these, and I am open to updating them if the authors can address my concerns. The paper presentation, especially the technical analysis, is not quite informative.

**Strengths:**

1. This is an interesting generalization of existing leverage score-based projection maintenance algorithms, which are used for some sota linear programming algorithms.
2. The algorithm is deterministic and has a good update time.
3. The algorithm also has differential privacy guarantees.

**Weaknesses:**

See questions below.

**Questions:**

1. It is unclear to me how this generalization can be used for important problems. Is there any easy way to see what the update times mean here?
2. I would also like to see how do these contrast with the corresponding result for leverage scores.
3. In this case, I would expect that the matrix A keeps changing more than just the weights which seemed to be more related to how IPM updates worked. Can that be captured by the framework?
4. Usually sketching matrices give additive error guarantees or errors that depend on the Frobenius norm of the matrix. How does this work manage multiplicative errors?
5. The main result for the update time analysis of the algorithm, Lemma 4.5 does not seem to have a proper proof. The authors just say that it is identical to Cohen et al. Is it exactly the same and which part of that proves this?

---

### Official Review · Reviewer_ZjAZ · 2025-11-05

**Soundness:** 2
**Presentation:** 1
**Contribution:** 2
**Rating:** 2
**Confidence:** 2

**Summary:**

The paper studies the question of estimating Lewis weights projection, i.e., maintain the projection
$$ W^{1/2-1/p} A(A^\top W^{1-2/p}A)^{-1}A^\top W^{1/2-1/p}$$

with and without privacy constraints.

They assume that the diagonal weight updates comes as a form of a stream.

**Strengths:**

The paper proposes an algorithm that maintains the projection and compute an approxima- tion of matrix-vector product between the projection matrix and any online vector, $h \in \mathbb R^n$.

**Weaknesses:**

The first 9 pages just read like algorithm and theorem and lemma statement. There is no exposition and no comparison with any related work. I would for example, like to see, what would happen when $p\to 2$ or settings where some results are already known. In particular, comparison with any previous prior work is definitely missing. I have no idea whether $n^2d^{\omega-2}$ is efficient time when it comes to initialization. In particular, initialization process is the most naive approach one would take. Similarly, it is not clear whether it is fine to have $n^{1+b+o(1)}$ or $n^{1+a+o(1)}$. On that front, what is $a$? Also, the update time proof is identical to previous works.


In particular, it is hard to parse what is the main contribution of the paper from the technical perspective? Or is it simply using previous works in a more general setting of Lewis weight?

I honestly do not think the paper is ready for acceptance.

**Questions:**

What are the main technical ideas in the paper when it comes to proof or algorithmic design?

---

### Official Review · Reviewer_nxut · 2025-11-10

**Soundness:** 4
**Presentation:** 2
**Contribution:** 3
**Rating:** 4
**Confidence:** 2

**Summary:**

This paper studies the problem of dynamically maintaining the projection $P(W)= W^{\frac{1}{2}-\frac{1}{p}} A\left(A^{\top} W^{\frac{1}{2}-\frac{1}{p}} A\right)^{-1} A^{\top} W^{\frac{1}{2}-\frac{1}{p}}$ under updates to the diagonal weight matrix $W$, which generalizes the classical dynamic projection maintenance setting used in interior-point methods (where $B=\sqrt{W} A$ ). The authors present a deterministic data structure that efficiently maintains an approximate projection, supporting fast Update and Query operations with sublinear amortized update time (Theorem 4.1; Algorithms 1-3). The approach combines the Woodbury identity, sketching techniques, and stability bounds to avoid recomputing matrix inverses from scratch. The paper further extends the framework to the $(\varepsilon, \delta)$-differentially private setting, providing formal privacy and utility guarantees. Overall, the work contributes a theoretical, more general dynamic projection-maintenance method with potential implications for fast and private optimization algorithms.

**Strengths:**

The paper offers a notable conceptual generalization of the dynamic projection maintenance problem by extending it from the classical $\ell_2$ setting (which is central to interior-point methods) to a broader $\ell_p$-Lewis-weight-based formulation. This unification is original as it bridges techniques from leverage-score sampling, Lewis weights, and adaptive preconditioning under one framework. The addition of a differentially private variant further broadens applicability to privacy-sensitive optimization, representing a meaningful extension of prior results.

The work demonstrates a high level of mathematical rigor. The proposed deterministic data structure is carefully designed, leveraging the Woodbury identity, sketching matrices, and stability-aware analysis to achieve sublinear amortized update time. The theoretical guarantees include detailed bounds for both correctness and runtime. The differential privacy extension is also handled with care, incorporating truncated Laplace mechanism and composition analysis to provide formal privacy and utility guarantees. Overall, the proofs appear sound and build logically on prior literature.

The paper addresses a problem of significance to optimization theory, particularly for fast interior-point methods, dynamic sampling, and private or adaptive optimization. If adopted, the proposed framework could reduce computation in iterative solvers and expand the set of algorithms that can be made dynamic and privacy-preserving. While the exposition is dense and may challenge readers not already familiar with the area, the core ideas are valuable and have the potential to influence subsequent work on efficient projection-based methods.

**Weaknesses:**

The paper is entirely theoretical and does not provide any empirical evaluation or implementation results. Given that the core motivation is to reduce computational cost in dynamic settings, experiments demonstrating runtime improvements over baseline recomputation or prior dynamic methods (Cohen 2021b) would be halpful. While the paper heavily relies on the theoretical tools such as projection maintenance, fast matrix multiplication etc., an empirical study would help validate whether the proposed amortized bounds translate into practical gains, especially when numerical issues, constant factors, and implementation overhead are taken into account. Even small-scale experiments, synthetic benchmarks, or a proof-of-concept implementation would strengthen the paper by clarifying its real-world feasibility and performance trade-offs.


While the paper is technically rigorous and complete in its theoretical development, I found the overall presentation is dense and may be challenging for readers who are not already familiar with projection maintenance, $\ell_p$-Lewis weights, or differential privacy. At times, the paper reads more like a sequence of lemmas, definitions, and technical proofs rather than a guided narrative that builds intuition. Strengthening the connective explanations (especially, Sections 3, 4 and 5) and clarifying how each component contributes to the final result (Theorem 4.1) would make the paper more accessible to a broader theoretical audience. The main exposition quickly moves into formal notation, invariants, and amortized runtime arguments, offering limited high-level intuition behind the design choices or the interaction between the $\ell_p$ formulation and projection updates. The Technical Overview in Section 5 focuses largely on stating the privacy and utility guarantees rather than conceptually motivating how the mechanism operates or why it works, and the transition from the classical $\sqrt{W} A$ setting to the generalized $W^{1 / 2-1 / p} A$ case could be better contextualized for clarity. Although the paper includes full algorithms and proofs, the narrative would benefit from stronger structuring, more intuitive guidance for the reader, and periodic explanatory commentary to bridge the gap between notation-heavy derivations and the underlying ideas.

**Questions:**

In context of the general weaknesses outined above, I've the following questions/comments for the authors:


1. Some of the DP-related lemmas appear redundant and could be consolidated for clarity. For example, Lemma 5.3 states the sensitivity of $J=W^{1 / 2-1 / p} A$, and Lemma 5.4 immediately applies a standard DP mechanism (via Lemma 3.4) using that sensitivity, restating nearly identical assumptions and definitions. Combining these into a single lemma or a lemma-corollary pair would reduce repetition and improve readability.

2. How difficult would it be to implement this data structure in practice?  Are there numerical stability issues when $p$ is far from $2$ (affects conditioning of $A^{\top} W^{1-2 / p} A$ )?

3. In Algorithm 2 (step 21), the Woodbury identity is used for efficient updates.  Could the authors show the explicit mapping from the general identity  $(M + U C V)^{-1}$ to their matrix update form to make this step fully transparent?

4. Theorem 4.1 references Lemmas 4.3-4.5 for the amortized cost.  Could the authors briefly explain how the stochastic smoothness conditions on $w^{(k)}$ (with constants $C_1, C_2$) translate into the stated runtime bound?

5. Section 5 provides separate differential privacy guarantees for each component (e.g., $\left(\varepsilon_J, \delta_J\right),\left(\varepsilon_H, \delta_H\right)$, etc.). Could the authors clarify how these are composed to yield an end-to-end $(\varepsilon, \delta)$ guarantee for the full mechanism? It would be helpful to explicitly state the composition rule used and the resulting overall privacy budget. In particular, is the composition handled solely via the basic composition result stated in Lemma 3.9, or is a tighter or alternative composition method applied?

---

### Meta-Review · Area_Chair_qUQd · 2026-01-04

**Summary:**

This paper is not appropriate for ICLR. The problem motivation is weakly explained and reviewers brought up the poor exposition and marginal technical contribution. I recommend reject, and that the authors significantly revise the presentation of the paper.

**Reviewer Concerns:**

N/A, there was no rebuttal.

**Reviewer Scores:**

N/A, there was no rebuttal.

---

### Decision · Program_Chairs · 2026-01-26

Reject